# Avian Sarcoma and Leukosis Virus Envelope Glycoproteins Evolve to Broaden Receptor Usage Under Pressure from Entry Competitors [note 1]

**DOI:** 10.3390/v11060519

**Published:** 2019-06-05

**Authors:** Audelia Munguia, Mark J. Federspiel

**Affiliations:** Department of Molecular Medicine, Mayo Clinic, Rochester, MN 55905, USA; amunguia@mmsholdings.com

**Keywords:** the subgroup A through E avian sarcoma and leukosis viruses, genetic selection inhibiting entry, receptor use expansion

## Abstract

The subgroup A through E avian sarcoma and leukosis viruses (ASLV(A) through ASLV(E)) are a group of highly related alpharetroviruses that have evolved their envelope glycoproteins to use different receptors to enable efficient virus entry due to host resistance and/or to expand host range. Previously, we demonstrated that ASLV(A) in the presence of a competitor to the subgroup A Tva receptor, SUA-rIgG immunoadhesin, evolved to use other receptor options. The selected mutant virus, RCASBP(A)Δ155–160, modestly expanded its use of the Tvb and Tvc receptors and possibly other cell surface proteins while maintaining the binding affinity to Tva. In this study, we further evolved the Δ155–160 virus with the genetic selection pressure of a soluble form of the Tva receptor that should force the loss of Tva binding affinity in the presence of the Δ155–160 mutation. Viable ASLVs were selected that acquired additional mutations in the Δ155–160 Env hypervariable regions that significantly broadened receptor usage to include Tvb and Tvc as well as retaining the use of Tva as a receptor determined by receptor interference assays. A similar deletion in the hr1 hypervariable region of the subgroup C ASLV glycoproteins evolved to broaden receptor usage when selected on Tvc-negative cells.

## 1. Introduction

All retroviruses initially synthesize their envelope glycoproteins as a precursor that is subsequently processed into two glycoproteins, the surface (SU) and transmembrane (TM) glycoproteins that then form a trimer of SU:TM heterodimers [1]. The SU glycoprotein contains the domains important for interaction with a host protein receptor. The subgroup A through E avian sarcoma and leukosis viruses (ASLV(A) through ASLV(E)) are a group of highly related alpharetroviruses that have evolved their *env* genes encoding the viral envelope glycoproteins from a common ancestor to use members of very different host protein families as receptors to enable efficient virus entry [2,3,4]. The evolution to use alternative receptors was presumably due to the development of host resistance and/or to expand host range. The ASLV(A) through ASLV(E) SU glycoproteins are highly conserved except for five hypervariable domains, vr1, vr2, hr1, hr2, and vr3 [5,6,7]. A variety of studies have identified hr1 and hr2 as the principle binding domains between the viral glycoprotein trimer and the host protein receptor, with vr3 contributing to the specificity of the receptor interaction for initiating efficient infection (Figure 1) [8,9,10,11,12,13].

Members of three very different families of proteins have been identified to be receptors of these five ASLVs, although all are simple, single-spanning membrane proteins. Tva proteins are related to low-density lipoprotein receptors (LDLR) and are receptors for ASLV(A) [14,15,16]. Tvb proteins are related to tumor necrosis factor receptors and are receptors for ASLV(B), ASLV(D), and ASLV(E) [17,18,19,20]. Tvc proteins are related to mammalian butyrophilins, are members of the immunoglobulin protein family, and are receptors for ASLV(C) [21,22]. The chicken ASLV receptor alleles have been characterized in multiple susceptible and resistant chicken strains [23]. A variety of mutations were identified that either result in a severely truncated or complete absence of the receptor protein, or point mutations often changing cysteine residues and altering receptor protein folding to reduce the binding affinity between the mutant receptor protein and the ASLV Env trimer [19,21,24,25,26]. These studies provide some examples of the natural receptor variants encountered by ASLVs that may have led to the evolution of receptor usage and the subgroup A–E ASLVs.

We have exploited genetic selection strategies to force the replication-competent ASLVs to naturally evolve and acquire mutations to escape the pressure on virus entry and yield a functional replicating virus [10,27,28,29]. This approach allows the simultaneous selection of multiple mutations in multiple functional domains of the envelope glycoprotein that may be required to yield a functional virus. One genetic selection strategy has taken advantage of the fact that secreted forms of the ASLV receptors potently bind the Env trimer to compete with cell associated receptors: The extracellular domains of the receptor fused to an IgG domain to form an immunoadhesin inhibitor [10,17,19,28]. A second strategy employs SU domain immunoadhesins to bind to the cellular receptor and effectively eliminate its availability to bind the virus envelope glycoproteins, thereby applying evolutionary pressure to adapt to use an alternative receptor [27]. Finally, there are a wide variety of chicken lines that effectively do not express certain functional ASLV subgroup receptors that can then be used to genetically select a virus with expanded receptor usage.

In order to understand how the ASLV subgroups may have evolved to use other cell surface proteins as receptors, presumably in a stepwise mechanism from an initial Env glycoprotein to the subgroup A to E subgroups, we have employed genetic selection strategies using replication-competent ASLV vectors, the RCASBP series [30,31]. Previously, we demonstrated that ASLV(A), in the presence of the receptor competitor SUA-rIgG immunoadhesin, evolved the use of other receptor options to circumvent the lack of Tva use to efficiently infect avian cells [27]. The selected mutant virus, RCASBP(A) with a deletion of 155–160 (Δ155–160) in the hr1 domain of the EnvA glycoprotein (Figure 1), modestly expanded its use of the Tvb and Tvc receptors and possibly other cell surface proteins while maintaining the binding affinity to Tva. In this study, we sought to further evolve the Δ155–160 virus with the genetic selection pressure of a soluble form of Tva, cksTva-mIgG immunoadhesin [15], that would force the loss of Tva binding affinity in the presence of the Δ155–160 mutation. Viable ASLVs were selected that acquired additional mutations in the Δ155–160 Env hypervariable regions that significantly broadened receptor usage to include Tvb and Tvc as well as retaining the use of Tva as a receptor determined by receptor interference assays. A similar deletion in the hr1 hypervariable region of the subgroup C ASLV glycoproteins evolved to broaden receptor usage when selected on Tvc-negative cells.

## 2. Materials and Methods

### 2.1. Soluble Receptor and Retroviral Vector Constructs

The construction of the gene-encoding chicken soluble Tva-mIgG immunadhesin (*ckstva-mIgG*), and the generation of a clonal DF-1 cell line that expresses a high level of the CKsTva-mIgG from the TFANEO transfection vector, TF/cksTva-19, were previously described [28]. The construction of the gene-encoding quail soluble Tva-mIgG immunadhesin (*qstva-mIgG*), and the generation of a clonal DF-1 cell line that expresses a high level of the QsTva-mIgG from the TFANEO transfection vector, TF/qsTva-4, were previously described [28]. Cleared supernatant harvested from DF-1 cells infected with RCASBP(A)*stvb^S3^-mIgG* was used as the source of sTvb^S3^-mIgG immunoadhesin protein [32]. Cleared supernatant harvested from DF-1 cells infected with RCASBP(A)*stvc-mIgG* was used as the source of sTvc-mIgG immunoadhesin protein [20].

The envelope subgroup A RCASBP(A), envelope subgroup B RCASBP(B), and envelope subgroup C RCASBP(C) ASLV retroviral vectors, as well as these vectors that also contain the heat stable human placental alkaline phosphatase gene (AP), RCASBP(A)AP, RCASBP(B)AP, and RCASBP(C)AP retroviral vectors, have been described previously [30,31]. The genetic selection and characterization of the RCASBP(A) Δ155–160 mutant virus was described previously [27].

### 2.2. Cell Culture and Virus Propagation

Chicken embryo fibroblasts (CEFs) were prepared from 10-day-old embryos of chicken lines Line C and Line 15I5. Line C cells do not express a functional Tva receptor but are susceptible to infection by subgroups B-E ASLVs [24]. Line 15I5 cells do not express a functional Tvc receptor but are susceptible to infection by subgroups A, B, D, and E ASLV [23]. The DF-1 cell line is a continuous fibroblastic cell line derived from Line 0 CEFs [33,34]. Line 0 cells are susceptible to infection by subgroups A, B, C, and D ASLVs. In addition, Line 0 cells do not contain endogenous retroviruses homologous to the subgroup A to E ASLVs. CEFs and DF-1 cells, a continuous fibroblastic cell line derived from Line 0 CEFs [33,34], were maintained in DMEM (Gibco/Invitrogen) supplemented with 10% fetal bovine serum (Gibco/Invitrogen), 100 units of penicillin per ml, and 100 µg of streptomycin per ml (Quality Biological, Inc, Gaithersburg, MD, USA), at 39 °C and 5% CO_2_. The TF/cksTva-19 cell line was maintained as the DF-1 cells but the medium supplemented with 250 µg/mL G418 as described previously [28].

Virus propagation was initiated in CEFs and DF-1 cells by calcium phosphate transfection of purified plasmid DNA (10 μg) that contained the retroviral vector in proviral form. The transfected cells were passaged, normally 1:3 or 1:6, when confluent to allow virus replication and spread. Viral spread was monitored by assaying culture supernatants for ASLV capsid (CA) protein by ELISA [35]. Virus stocks were generated from the cell supernatants by clearing the cellular debris by centrifugation at 2000× *g* for 10 min at 4 °C and stored in aliquots at −80 °C. In some experiments, cleared infected-cell supernatants were used to infected fresh, uninfected cells, to further select and/or characterize virus replication of evolved mutant virus pools.

### 2.3. Infectious ASLV Alkaline Phosphatase Assays

In AP titer assays, cell cultures (~30% confluent) were incubated with 10-fold serial dilutions of the RCASBP/AP virus stocks for 42–48 h at 39 °C. The assay for alkaline phosphatase activity was described previously [28].

### 2.4. Cloning and Nucleotide Sequence Analysis of Integrated Viral DNA 

DNA was isolated from infected cells in culture using the QIAamp Tissue Kit (Qiagen, Hilden, Germany). The entire *env* gene regions were amplified by PCR, the amplified products were separated by agarose gel electrophoresis, and the ~2.0 kb product purified and cloned into pCR2.1-TOPO using the TOPO TA Cloning kit (Invitrogen, Carlsbad, CA, USA). The nucleotide sequence of the *env* genes were determined by the Mayo Clinic Molecular Biology Core on a Perkin Elmer ABI PRISM 377 DNA sequencer (with XL upgrade) with PE Applied Biosystems ABI PRISM dRhodamine Terminator Cycle Sequencing Ready Reaction Kit and AmpliTaq DNA Polymerase (PE Applied Biosystems, Foster City, CA, USA).

### 2.5. Flourescence-Activated Cell Sorting (FACS) Analysis of Envelope Glycoprotein Binding to Receptor

Uninfected DF-1 cells or DF-1 cells infected with either wild-type or mutant ASLV viruses were removed from culture with Trypsin de Larco (Quality Biological, Inc.) and washed with Dulbecco’s phosphate buffered saline (PBS). The cells were fixed with 4% paraformaldehyde in PBS at room temperature for 15 min and then washed with PBS. Approximately 1 × 10^6^ cells in PBS supplemented with 1% calf serum (PBS-CS) were incubated with supernatant containing either the CKsTva-mIgG, QsTva-mIgG, sTvb^S3^-mIgG, or sTvc-mIgG on ice for 30 min. The cells were then washed with PBS-CS and incubated with 5 µL of goat anti-mouse IgG (H+L) linked to phycoerythrin (Kirkegaard and Perry Laboratories, Gaithersburg, MD, USA) in PBS-CS (1 mL total volume) on ice for 30 min. The cell:soluble receptor-mIgG:Ig-phycoerythrin complexes were washed with PBS-CS, resuspended in 0.5 mL PBS-CS, and analyzed with a Becton Dickenson FACSCalibur using CELLQuest 3.1 software.

## 3. Results and Discussion

Two different mutant virus pools were selected by the RCASBP(A)Δ155–160 virus to escape the cksTva-mIgG immunadhesin entry inhibitor. The RCASBP(A)Δ155–160 mutant virus evolved under the selective pressure of SUA-rIgG immunadhesin that reduced the expression of the Tva receptor on the cell surface, and if Tva was displayed, competed with the subgroup A envelope glycoproteins on the virion for binding the cell surface Tva receptor, forcing RCASBP(A) to alter its receptor usage to enter cells. However, the RCASBP(A)Δ155–160 envelope glycoproteins still maintained wild-type levels of binding affinity for Tva [27]. We sought to further evolve the receptor usage of the Δ155–160 virus with the genetic selection pressure of a soluble form of the chicken Tva receptor, cksTva-mIgG immunoadhesin, which would presumably select mutations in the Δ155–160 envelope glycoproteins that would alter binding affinity to Tva while in the presence of the Δ155–160 mutation.

The TF/cksTva-19 cell line, a DF-1 clonal line that expresses high levels of the chicken cksTva-mIgG immunoadhesin inhibitor, blocks virus entry by binding the subgroup A envelope glycoproteins on the virions and blocking access to cell surface Tva receptors. TF/cksTva-19 cells were infected with RCASBP(A)Δ155–160 (10 mL supernatant; ~10^5^ ifu/mL; ~0.33 M.O.I.) either directly (Δ155–160) or preincubated with cksTva-mIgG (Δ155–160+sTva1) to increase the initial inhibitory effect by binding to virions (Figure 2A). The specificity of the cksTva-mIgG immunadhesin is to subgroup A ASLV envelope glycoproteins, with no significant inhibition of the entry of virions with other ASLV envelope glycoprotein subgroups [28]. TF/cksTva-19 cells were also infected with a control subgroup B ASLV, RCASBP(B), that replicated well as expected, producing high levels of CA protein a week after infection. The Δ155–160 and Δ155–160+sTva1 infected cultures both experienced a significant delay to day 26 before high levels of virus were produced.

In a second round of selection, TF/cksTva-19 cells were infected with day 32 supernatants (~1.0 mL) from the Δ155–160 and Δ155–160+sTva1 cultures to further define the most efficient and resistant evolved viruses. A third infection was also performed using Δ155–160+sTva1 supernatant preincubated with cksTva-mIgG (Δ155–160+sTva2) to increase the inhibitory effect prior to exposure to cells. The virus pools from all three infected cultures replicated well after a short delay reaching high CA levels by day 12 (Figure 2B). Infected cells were harvested from the three cultures, genomic DNA isolated, and the integrated ASLV *env* genes amplified using PCR, cloned, and the nucleotide sequences determined (Figure 2C). All virus *env* gene clones (8/8) analyzed from the Δ155–160 culture contained the original Δ155–160 deletion but acquired a single mutation in the vr3 hypervariable region, G268E. The *env* genes of the mutant virus pools in the Δ155–160+sTva1 and Δ155–160+sTva2 cultures also retained the Δ155–160 deletion, and all had acquired the G133D mutation in hypervariable region hr1, but several additional mutations in the hr1 hypervariable region were also acquired. The Δ155–160+sTva1 culture was an ~equal mixture of two variant viruses: G133D+L143P (4/8) and G133D+Y142H (3/8). The Δ155–160+sTva2 culture contained the same two variant viruses but now with the predominant clone being G133D+Y142H (6/8).

Four new recombinant mutant RCASBP(A)Δ155–160 viruses were constructed to further characterize the phenotypes of the additional selected mutations: RCASBP(A)Δ+G133D; RCASBP(A)Δ+G133D+Y142H; RCASBP(A)Δ +G133D+L143P; and RCASBP(A)Δ+G268E. DF-1 cells were transfected with plasmids encoding the four new mutant viruses, the original RCASBP(A)Δ155–160 mutant virus, and the RCASBP(A) wild-type virus. The transfected cell cultures were passaged when confluent to allow virus replication and spread (Figure 3A). All six viruses replicated well on DF-1 cells (Figure 3A) but produced different maximum infectious titers (Figure 4A): Wild-type RCASBP(A) reached a titer of ~10^6^ ifu/mL and RCASBP(A)Δ155–160 reached a maximum titer of ~10^5^ ifu/mL as expected from previous work [27]. Subgroup A RCASBP(A) produced the highest titers in DF-1 cells without causing any observable cytotoxic effects to infected cells. However, the replication of subgroup B, C, and D RCASBP vectors often caused a cytotoxic crisis to a percentage of infected cells resulting in the loss of ~10-fold infectious titer compared to RCASBP(A) [11,36]. The changes in receptor usage of RCASBP(A)Δ155–160 to evade the original SUA-rIgG immunadhesin inhibition resulted in viruses that produce titers similar to non-subgroup A ASLVs. All four newly selected viruses replicated well in DF-1 cells but only reached a maximum infectious titer of ~10^4^ ifu/mL; ~10-fold lower than the original RCASBP(A)Δ155–160 but without displaying obvious cytotoxicity.

TF/cksTva-19 cells were also transfected with plasmids encoding the four new mutant viruses, the original RCASBP(A)Δ155–160 mutant virus, the RCASBP(A) wild-type virus, and the RCASBP(B) virus that should not be inhibited by the cksTva-mIgG immunoadhesin. The transfected cell cultures were passaged when confluent to allow virus replication and spread (Figure 3B). As expected, RCASBP(B) replicated well with no growth lag. Also, as expected, both the replication of the wild-type RCASBP(A) and the RCASBP(A)Δ155–160 mutant viruses were significantly inhibited in the presence of the cksTva-mIgG immunadhesin inhibitor. The three new mutant viruses, RCASBP(A)Δ+G133D+Y142H, RCASBP(A)Δ+G133D+L143P, and RCASBP(A)Δ+G268E, replicated well in the presence of the cksTva-mIgG immunadhesin inhibitor demonstrating that these mutations provide a growth advantage under these conditions. However, while all 16 clones from the Δ155–160+sTva1 and Δ155–160+sTva2 round 2 selections contained the G133D mutation in hr1, the virus with this mutation alone does not replicate as well. These results were verified by performing a growth curve using TF/cksTva-19 cells infected with each virus at an M.O.I. of 0.01 (Figure 3C): RCASBP(A)Δ+G133D+Y142H, RCASBP(A)Δ+G133D+L143P, and RCASBP(A)Δ+G268E replicated well, demonstrating that these mutations provide a growth advantage under these conditions.

### 3.1. The Additional Mutations Acquired by the RCASBP(A)Δ155–160 Alter Receptor Usage 

We expected the selective pressure of the cksTva-mIgG immunoadhesin to cause the virus to evolve the Δ155–160 subgroup A envelope glycoproteins to reduce binding affinity for the sTva inhibitor, thus altering the new mutant receptor usage. A series of receptor interference assays were performed to assess changes in receptor usage in chicken cells. Uninfected DF-1 cells and DF-1 cells chronically infected with ASLVs of subgroup A (RCASBP(A)), subgroup B (RCASBP(B)), subgroup C (RCASBP(C)), or subgroup J (HPRS103) were challenged with 10-fold serial dilutions of virus stocks produced using DF-1 cell of wild-type RCASBP(A), RCASBP(B), and RCASBP(C) control viruses, the original RCASBP(A)Δ155–160 mutant virus, and the four new mutant viruses with the viral titers determined by AP infectious titer assay (Figure 4A). As expected, the wild-type RCASBP viruses efficiently infected uninfected DF-1 cells, and DF-1 cells infected by non-identical subgroup ASLVs, while DF-1 cells already infected with a subgroup ASLV inhibited re-infection by the identical subgroup by 5–6 logs due to receptor interference by the cells continuously expressing that subgroup envelope glycoproteins. The receptor interference is exquisitely specific for that subgroup ASLVs demonstrating that the subgroup A to E ASLVs each evolved to use one specific receptor to efficiently infect avian cells.

As reported previously, the RCASBP(A)Δ155–160 mutant virus could now infect the subgroup A infected DF-1 cells more efficiently than wild-type RCASBP(A), and was somewhat less efficient at infecting subgroups B and C infected DF-1 cells compared to wild-type (5–10-fold), which may indicate the RCASBP(A)Δ155–160 virus can use the Tvb and Tvc receptors to enter cells. The additional mutations acquired by the RCASBP(A)Δ155–160 virus in this study all increased the virus’ ability to infect subgroup A infected DF-1 cells compared to uninfected cells by >10-fold (Figure 4A). Evolving to include additional mutations in the background of the Δ155–160 deletion resulted in viruses even less efficient at infecting subgroups B and C infected cells indicating more efficient use of the Tvb and Tvc receptors compared to RCASBP(A)Δ155–160. Compared to RCASBP(A)Δ155–160, the RCASBP(A)Δ+G133D virus was ~10-fold less efficient infecting subgroups B and C infected cells; RCASBP(A)Δ+G133D+Y142H and RCASBP(A)Δ+G133D+L143P viruses were ~100-fold less efficient infecting subgroups B and C; while the RCASBP(A)Δ+G268E virus was ~100-fold less efficient infecting subgroup B and ~1000-fold less efficient infecting subgroup C infected cells.

The viruses were also analyzed for their efficiency to infect Line C CEFs that lack a functional subgroup A ASLV Tva receptor (Figure 4B). The infection of Line C CEFs by wild-type RCASBP(A) is reduced ~3-logs; the RCASBP(A)Δ155–160 mutant virus infects Line C more efficiently with only a ~10-fold reduction in titer, again indicating that the Δ155–160 deletion expands the receptor usage. The additional mutations acquired by the RCASBP(A)Δ+G133D+Y142H, RCASBP(A)Δ+G133D+L143P and RCASBP(A)Δ+G268E viruses result in the entry of these viruses not requiring Tva since their titers are the same on both DF-1 and Line C CEFs. The titer of the RCASBP(A)Δ+G133D virus was reduced by a similar level as the RCASBP(A)Δ155–160 mutant virus.

The viruses were also analyzed for their efficiency to infect Line 15I5 CEFs that lack a functional subgroup C ASLV Tvc receptor (Figure 4C). The lack of a functional Tvc receptor reduces the ability of wild-type RCASBP(C) to infect Line 15I5 CEFs resulting in a 1-2-log lower titer; since Line 15I5 has a Tva receptor, subgroup A ASLVs infect efficiently with titers at similar levels to DF-1 cells. It is interesting that the titer of the RCASBP(A)Δ+G133D virus was reduced by a similar level as the wild-type RCASBP(C) using Line 15I5 CEFs perhaps indicating that the G133D mutation reduces the efficiency of entry in general with any receptor.

### 3.2. The Mutations Acquired by the RCASBP(A)Δ155–160 Mutant Virus Significantly Reduce Binding Affinity for the Tva Receptor

We expected that the evolutionary pressure of the cksTva-mIgG immunoadhesin would select for mutations in the ASLV glycoprotein that would reduce the binding affinity of the glycoprotein to this inhibitor and thereby enable the interaction with a different receptor for entry. DF-1 cells chronically infected with the ASLVs were used in a FACS-based assay to estimate changes in binding affinities using Tva receptor immunoadhesins: Chicken sTva CKsTva-mIgG (Figure 5A) and quail sTva QsTva-mIgG (Figure 5C). All six-virus infected DF-1 cell cultures expressed similar levels of envelope glycoprotein on the surface of the chronically infected cells. As reported previously, the RCASBP(A)Δ155–160 mutant envelope glycoproteins bound both the chicken and quail sTva-mIgG immunoadhesins with similar affinities as wild-type RCASBP(A) glycoproteins [27]. All four new mutant virus envelope glycoproteins bound both chicken and quail sTva-mIgG at significantly lower levels compared to wild-type and RCASBP(A)Δ155–160 glycoproteins: The new mutant glycoproteins may bind low levels of chicken sTva-mIgG while we could not detect any binding within the sTva-mIgG concentrations tested for the quail sTva-mIgG.

We and others have shown that binding of ASLV virus stocks with an appropriate receptor immunoadhesin significantly reduces the resulting titer from the antiviral effect of binding the viral glycoproteins. As another measure of receptor binding affinity, the titers of all six virus stocks were determined using parental DF-1 cells of the stock directly, or after each 10-fold serial dilution was first preabsorbed with the same amount of chicken sTva-mIgG (Figure 5B) or quail sTva-mIgG (Figure 5D). As expected, both the pre-absorption of chicken and quail sTva-mIgG reduced the titer of wild-type RCASBP(A) by greater than 2-logs and the RCASBP(A)Δ155–160 mutant virus by ~45-fold. All four new mutant viruses were still inhibited with chicken sTva-mIgG but at lower levels compared to RCASBP(A)Δ155–160 mutant virus supporting that the glycoprotein mutations have a lower binding affinity to chicken sTva-mIgG. For three of the new viruses, the results were similar, with a lower level of inhibition after pre-absorption with quail sTva-mIgG. One mutant virus, Δ+G133D+Y142H, was not inhibited at all after pre-absorption with quail sTva-mIgG. We have previously shown RCASBP(A) variants could be selected under pressure of the quail sTva-mIgG immunoadhesin evolved their glycoproteins to be able to distinguish the subtle differences between the chicken and quail Tva receptor homologs [10,15,28]. In this case, mutant RCASBP(A) contained mutations in the hr1 region that retained the wild-type level of binding affinity for the chicken Tva receptor, while eliminating the use of the quail Tva receptor. Several mutations were selected in the same region of Env A hr1, with one of the primary mutations, Y142N, occurring at the same Y142 residue as one of the mutants selected in this study.

Since the receptor interference patterns of the evolved RCASBP(A)Δ155–160 mutant viruses selected in this study indicate some alterations in receptor usage including possible use of Tvb, RCASBP(B) and the five RCASBP(A)Δ155–160 mutant viruses were assayed for their ability to bind to (Figure 5E) and susceptibility to the antiviral effects (Figure 5F) of the sTvb-mIgG immunoadhesin. Wild-type RCASBP(B) viral glycoproteins efficiently bound sTvb-mIgG while RCASBP(A)Δ155–160 mutant viruses did not detectably bind except for the possible very low binding of the RCASBP(A)Δ+G133D+L143P virus glycoproteins in the range of immunoadhesin concentrations tested (Figure 5E). The wild-type RCASBP(B) virus was susceptible to the antiviral effect of sTvb-mIgG pre-absorption resulting in a >3-log reduction in infectious titer (Figure 5F). While subgroup A viruses are not expected to bind at high affinities to nor be susceptible to the antiviral effect of sTvb-mIgG, as shown with the results of RCASBP(A)Δ155–160, two of the selected mutants are susceptible to a low but significant level of antiviral effect of sTvb-mIgG on titer, RCASBP(A)Δ+G133D+L143P (*P* = 0.0035)and RCASBP(A)Δ+G268E (*P* = 0.015) viruses, and a low level of binding affinity was detected for RCASBP(A)Δ+G133D+L143P.

### 3.3. RCASBP(C) Envelope Glycoproteins Evolved to Broaden Receptor Usage after Selection Using Tvc Receptor Negative Line 15I5 CEFs

This study has shown that under selective pressure blocking virus entry, a subgroup A ASLV could evolve to broaden receptor usage to other cell surface proteins including Tvb and Tvc. Since subgroup A and subgroup C ASLV envelope glycoproteins are closest in homology (Figure 1), we set out to evolve RCASBP(C) to be able to efficiently replicate in cells lacking a functional Tvc receptor, Line 15I5. We hypothesize that this pressure on virus entry would select envelop glycoprotein mutations that allow the subgroup C ASLV to broaden receptor usage to other proteins besides Tvc and possibly to use Tva.

Line 15I5 CEFs were infected with three different amounts of RCASBP(C), 0.1, 1.0, and 10 mL of 10^5^ ifu/mL stock, (multiplicity of infection, M.O.I., of 0.0033, 0.033, and 0.33, respectively) and passaged to allow virus replication, evolution, and spread through the culture (Figure 6A). A Line 15I5 CEF culture was infected with RCASBP(A) as a positive control for ASLV replication as this line expresses Tva. After 32 days, ASLV replication was detected only in the CEFs infected with 10 mL of RCASBP(C). A fresh culture of Line 15I5 CEFs was infected with 1.0 mL of day-37 RCASBP(C)-10 mL supernatant from the first round of selection, the cells passaged to allow virus replication, and spread (Figure 6B). Now, ASLV replication was detected by day 22. A third round of selection was done using second round day-25 RCASBP(C)-10 mL supernatant and a new infection of wild-type RCASBP(C) as a negative control (Figure 6C). This third round of selection evolved an ASLV that detectably replicates by day 10. Cells were harvested from third round day-18 RCASBP(C)-10 mL culture, genomic DNA isolated, and the ASLV *env* genes PCR amplified, cloned, and the ASLV *env* gene nucleotide sequences determined from 10 clones (Figure 6D). All 10 clones had a 20-amino-acid deletion, Δ144–160, in the hr1 hypervariable region of EnvC glycoprotein; 2/10 clones also contained an additional mutation F142S, Δ144–160+F142S.

Recombinant RCASBP(C) viruses were constructed with the Δ144–160 deletion, the Δ144–160+F142S mutations, and only the F142S mutation to assess the phenotype of this mutation alone. DF-1 cells were transfected with plasmid DNA encoding the wild-type RCASBP(C) or one of the three new recombinant mutant viruses, passaged to allow virus replication and spread, and virus supernatants collected as stocks. Unexpectantly, the RCASBP(C)Δ144–160 virus did not replicate while the wild-type RCASBP(C), RCASBP(C)Δ144–160+F142S, and the RCASBP(C)+F142S viruses replicated reasonably well (data not shown). As expected, the wild-type RCASBP(C) virus produced a titer of ~10^5^ ifu/mL using DF-1 cells; the RCASBP(C)+F142S virus titer was also ~10^5^ ifu/mL, while the titer of RCASBP(C)Δ144–160+F142S was ~10-fold lower, ~10^4^ ifu/mL. Since the RCASBP(C)Δ144–160+F142S mutations significantly reduced the binding affinity of the mutant glycoproteins for sTvc-mIgG compared to wild-type subgroup C glycoproteins, but the RCASBP(C)F142S mutation alone did not alter binding affinity, we assume the Δ144–160 mutation was primarily responsible for the loss of sTvc-mIgG binding affinity.

To characterize mutant glycoproteins possibly change(s) in receptor usage, the virus stocks were analyzed by receptor interference assay (Figure 6E). The wild-type RCASBP(C) produced a receptor interference pattern typical of a subgroup C ASLV: The titer determined using DF-1 cells previously infected with ASLV(C) was inhibited by >3-logs; the titer using DF-1 cells previously infected with subgroup B ASLV was inhibited 2–4-fold; while titers using DF-1 cells previously infected by subgroup A or J ASLV were the same as uninfected DF-1 cells. The receptor interference pattern of RCASBP(C)+F142S was similar to wild-type RCASBP(C). However, the titer of RCASBP(C)Δ144–160+F142S virus was significantly inhibited by ~100-fold using DF-1 cells previously infected with ASLV(C) rather than >3-logs, also inhibited by ~100-fold using DF-1 ASLV(B) cells, and finally ~10-fold inhibited using DF-1 ASLV(A) cells. As we observed for the EnvA Δ155–160 mutants (Figure 4A), the EnvC Δ144-160+F142S mutations significantly broadened receptor usage to indicate possible use of the Tva and Tvb receptors for virus entry. Since the RCASBP(C)Δ144–160 virus did not replicate in DF-1 cells, the F142S mutation rescues the ability of the Δ144–160 virus to replicate. The mixed mutant population analyzed from Line 15I5 Rd3 (Figure 6C,D) may have still been evolving toward a population of only the viable RCASBP(C)Δ144-160+F142S mutant.

## 4. Conclusions

The evolution of the subgroup A to E ASLVs to use alternative receptors for entry was presumably the result of the stepwise acquisition of mutations in the *env* gene that provided an escape to the lack of a suitable receptor caused by receptor polymorphisms and/or host range restrictions. The replication-competent ASLV viruses, the RCAS vectors, continue to provide a powerful system to experimentally force these viruses to evolve to escape blocks to entry.

Two evolutionary themes have formed from multiple studies that employed different blocks to ASLV entry. When ASLV confronts avian cells that lack the appropriate functional receptor, ASLV(A) with its extreme specificity for Tva, for example, an escape ASLV mutant virus is selected with a deletion in the hr1 region of the SU glycoprotein that expands the ability to use other cell surface proteins as receptors. The deletions selected map to a similar region of hr1 of viruses of all three receptor groups, Tva, Tvb, and Tvc (Figure 7): The original subgroup A ASLV(A)Δ155–160 mutant [27], the subgroup C ASLV(C)Δ144-160+F142S mutant reported here (Figure 6), and the recently reported subgroup B ASLV(B)Δ136–142 [32]. These hr1 deletions did not alter the binding affinity to the parental receptor in the ASLV(A)Δ155–160 and ASLV(B)Δ136–142 mutants; both of these mutations retained wild-type binding affinities for Tva and Tvb, respectively. Upon further selection with receptor immunoadhesin inhibitors that bind the viral glycoproteins directly, additional viable mutations evolved to significantly reduce the binding affinity of the viral glycoproteins to the parental receptor (Figure 3 and Figure 4). Mutations that altered receptor binding affinity clustered mainly in hr1 with some occurring in the vr3 region of the SU glycoprotein (Figure 7) [28,29,32]. This study supports the model that compensatory envelope glycoprotein mutations can accumulate in a stepwise manner to evade multiple blocks to virus entry and replication.

## Figures and Tables

**Figure 1 viruses-11-00519-f001:**
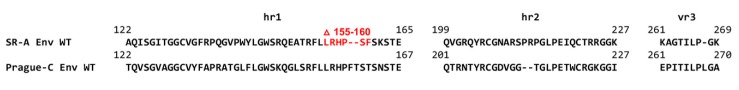
Comparison of the subgroup A and subgroup C avian sarcoma and leukosis viruses (ASLV) envelope glycoprotein hr1, hr2, and vr3 hypervariable regions. The protein sequence alignments were done using the ClustalW program in MacVector 14.5.3; gaps in the alignment are denoted by (−). The RCASBP(A) vector contains the envelope glycoprotein hypervariable regions from the Schmidt-Ruppin A subgroup A ASLV strain (SR-A Env WT) UniProt P03397; the RCASBP(C) vector contains the envelope glycoprotein hypervariable regions from the Prague-C subgroup C ASLV (Prague C Env WT) Genbank AAB59934.1. The RCASBP(A) Δ155–160 six residue deletion is highlighted in red.

**Figure 2 viruses-11-00519-f002:**
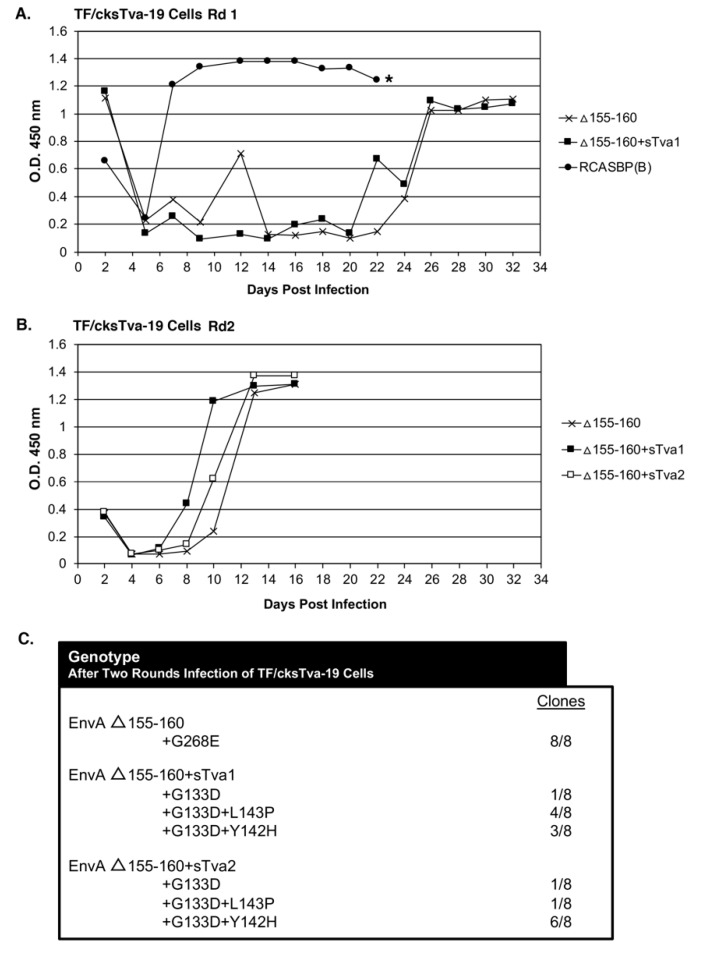
Evolution of RCASBP(A) Δ155–160 virus variants resistant to cksTva-mIgG. (**A**) TF/cksTva-19 cells expressing high levels of chicken sTva-mIgG protein were infected with DF-1 infection supernatant (10 mL) containing RCASBP(A) Δ155–160 virus (~10^5^ ifu/mL) either neat (Δ155–160) or after preincubation with cksTva-mIgG containing supernatant (Δ155–160+sTva1), or supernatant (1.0 mL) containing the subgroup B ASLV, RCASBP(B), and the infected cell cultures passaged to allow virus replication and spread. Viral growth was monitored by ELISA for the ASLV CA protein. The day the Δ155–160 infected culture began a transient period of ASLV induced cytotoxicity is marked with an asterisk (*). (**B**) The selected virus pools (1.0 mL) from the day 32 supernatants above were re-passaged in TF/cksTva-19 cells either neat (Δ155–160+sTva1), the selected supernatant preincubated with cksTva-mIgG supernatant (Δ155–160+sTva2), and non-selected wild-type RCASBP(A) Δ155–160 as a control, and the infected cell cultures passaged to allow virus replication and spread. (**C**) The ASLV(A) *env* nucleotide sequences were determined from clones generated from PCR amplified ASLV(A) *env* sequences from DNA isolated from infected cells. Rd: Round.

**Figure 3 viruses-11-00519-f003:**
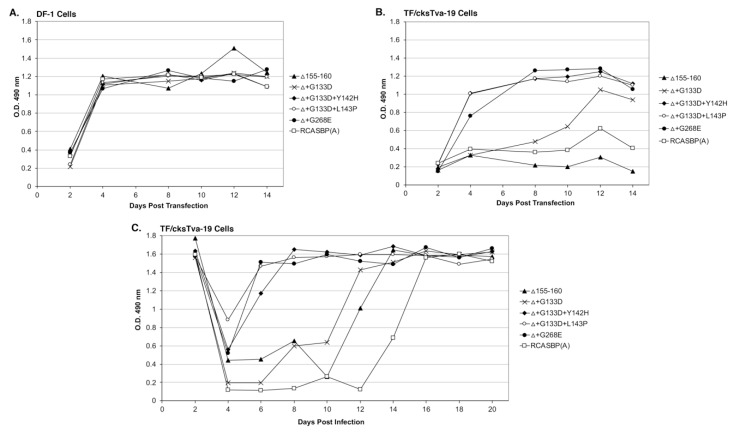
Virus replication rates of Δ155–160 and the evolved Δ155–160 mutant viruses. Viral growth was monitored by ELISA for the ASLV CA protein. Plasmids encoding the ASLVs were transfected into parental DF-1 cells (**A**) and TF/cksTva-19 cells (**B**), and the transfected cell cultures passaged to allow virus replication. (**C**) TF/cksTva-19 cells were infected with Δ155–160 and Δ155–160 mutant viruses at a multiplicity of infection of 0.01. The infected cell cultures were passaged to allow virus replication.

**Figure 4 viruses-11-00519-f004:**
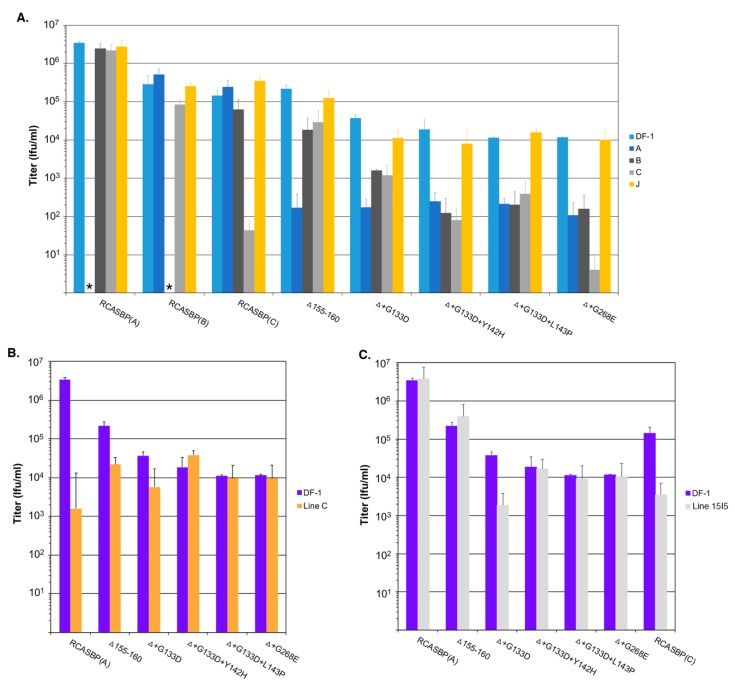
Analysis of receptor usage of the parental Δ155–160 and Δ155–160 mutant viruses. Infectious titers were determined using 10-fold serial dilutions of wild-type RCASBP(A), RCASBP(B), and RCASBP(C) viruses, the parental Δ155–160 virus, and the Δ155–160 mutant virus supernatants produced using DF-1 cells. The infectious titer was determined by the AP assay. No infectious units detected are denoted with (∗). The results shown are an average of three different experiments; error bars show standard deviations. (**A**) ASLV receptor interference patterns of the ASLVs infecting parental DF-1 cells, and DF-1 cells chronically infected with ASLV(A), ASLV(B), ASLV(C), or subgroup J ASLV, HPRS103. (**B**) The infectious titers of the ASLV viruses were determined using Line C CEFs that do not express a functional Tva receptor. (**C**) The infectious titers of the ASLV viruses were determined using Line 15I5 CEFs that do not express a functional Tvc receptor.

**Figure 5 viruses-11-00519-f005:**
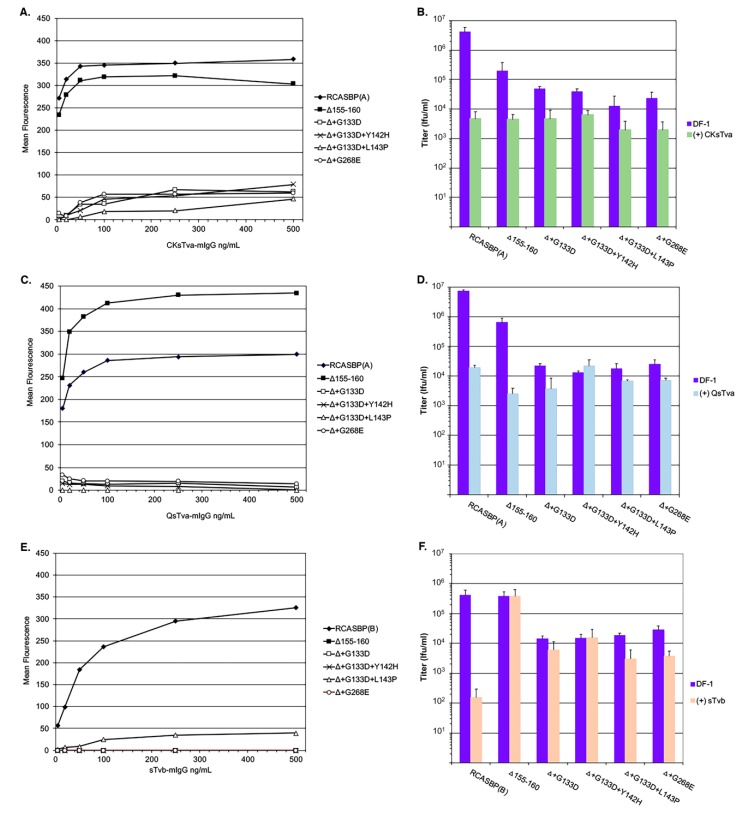
Analysis of the binding affinities and antiviral effects of receptor immunoadhesins on the evolved RCASBP(A) Δ155–160 virus variants. DF-1 cells infected with either control and variant ASLVs were fixed and incubated with different amounts of either the CKsTva-mIgG (**A**), QsTva-mIgG (**C**), or sTvb-mIgG (**E**) soluble ASLV receptor immunoadhesins and the envelope glycoprotein:soluble receptor complexes bound to sheep anti-mouse Ig antibody linked to fluorescein. The levels of fluorescein bound to the infected cells were quantitated by fluorescence-activated cell sorting (FACS) and the mean level of fluorescence was plotted as a function of soluble ASLV receptor concentration. Representative experiments are shown. Infectious titers were determined using 10-fold serial dilutions of control and variant ASLVs supernatants directly on DF-1 cells, or after each dilution was preabsorbed with a set amount of CKsTva-mIgG (**B**), QsTva-mIgG (**D**), or sTvb-mIgG (**F**) to allow the soluble receptor time to bind and inhibit infection of DF-1 cells. The infectious titer was determined by the AP assay.

**Figure 6 viruses-11-00519-f006:**
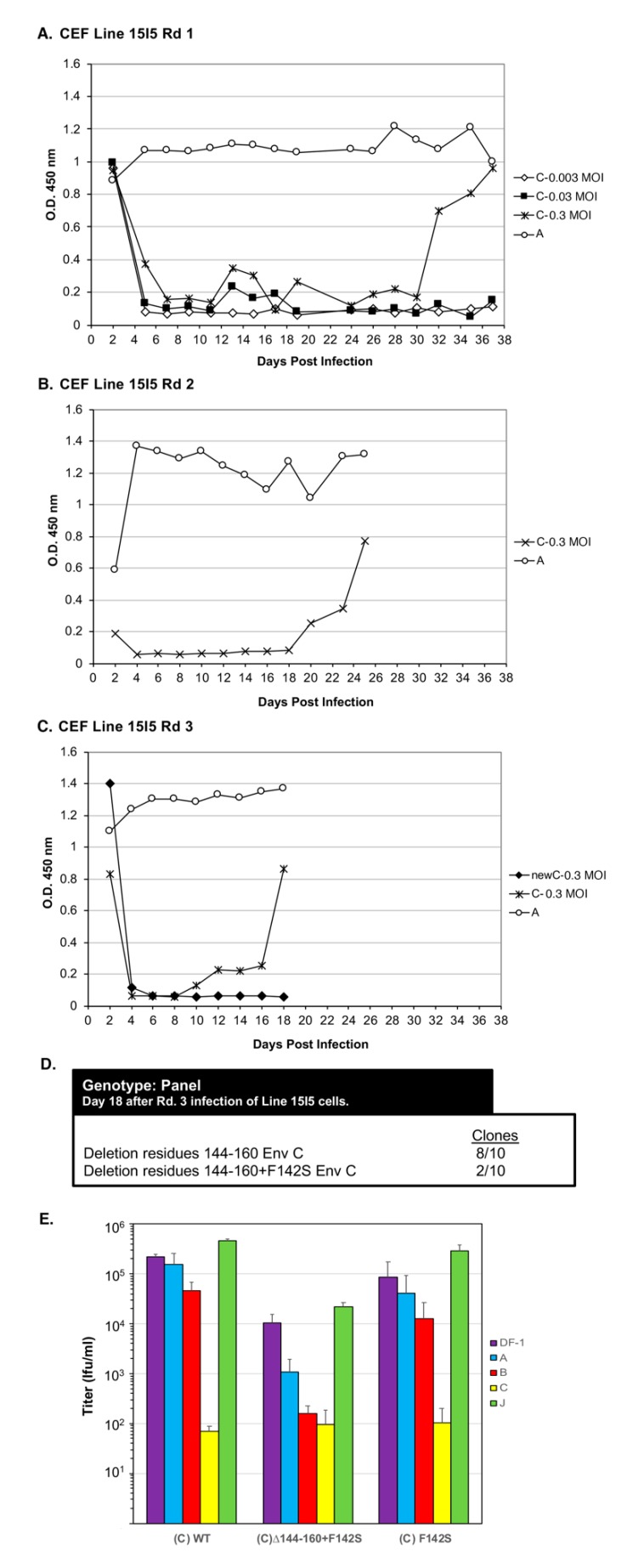
Genetic selection of RCASBP(C) mutants capable of efficient replication using Line 15I5 CEFs that lack a functional Tvc receptor. (**A**) Chicken embryo fibroblasts (CEFs) derived from Line 15I5 were infected with RCASBP(C) supernatant at 0.003, 0.03, and 0.3 M.O.I., or the positive control RCASBP(A) supernatant (0.1 mL), produced using DF-1 cells. The infected cells were passaged to allow virus replication. Viral growth was monitored by ELISA for the ASLV CA protein. (**B**) Supernatant (1.0 mL) harvested from the RCASBP(C) 0.3 M.O.I. infected culture on day 37, and supernatant (0.1 mL) harvested from day 37 RCASBP(A) culture, were used to infect fresh Line 15I5 CEFs in a second round of genetic selection. (**C**) In a third round of genetic selection, second-round day-25 supernatants from RCASBP(C) 0.3 M.O.I. and RCASBP(A) cultures were again used to infect fresh Line 15I5 CEFs; unselected RCASBP(C) virus was used as a “negative” control. (**D**) Cells were harvested from Rd 3 RCASBP(C) 0.3 M.O.I. infected culture, and clones were generated from PCR amplified ASLV(A) *env* sequences from isolated DNA from infected cells, and the ASLV(A) *env* nucleotide sequences determined. (**E**) ASLV receptor interference patterns of the ASLVs infecting parental DF-1 cells, and DF-1 cells chronically infected with ASLV(A), ASLV(B), ASLV(C), or subgroup J ASLV, HPRS103, were determined as described in Figure 4 legend. Rd: Round.

**Figure 7 viruses-11-00519-f007:**
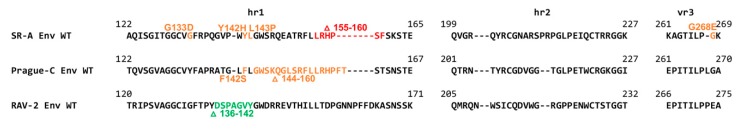
Comparison of the subgroup A and subgroup C ASLV envelope glycoprotein hr1, hr2, and vr3 hypervariable regions. The protein sequence alignments were done using the ClustalW program in MacVector 14.5.3; gaps in the alignment are denoted by (-). The RCASBP(A) vector contains the envelope glycoprotein hypervariable regions from the Schmidt-Ruppin A subgroup A ASLV strain (SR-A Env WT) UniProt P03397; the RCASBP(C) vector contains the envelope glycoprotein hypervariable regions from the Prague-C subgroup C ASLV (Prague C Env WT) Genbank AAB59934.1. The mutations selected and characterized in this study are highlighted in orange; the original RCASBP(A) Δ155–160 mutation is highlighted in red. The results from this study are compared to the ASLV(B) hr1, hr2, and vr3 regions of Rous associated virus-2 (RAV-2) Genbank AAA87241. Selection of a mutant RAV-2 that was able to infect avian cells that lacked a Tvb receptor had a deletion in hr1 residues 136–142 (Δ136–142) highlighted in green [32].

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
