# Peer review of "Avian Sarcoma and Leukosis Virus Envelope Glycoproteins Evolve to Broaden Receptor Usage Under Pressure from Entry Competitors [Author-notes fn1-viruses-11-00519]"

_viruses, 2019, doi:10.3390/v11060519_

Reviewer 1 Report

Munguia and Federspiel

The authors are examining the determinants of retroviral host range using the ASLV system. The interaction of the viral SU protein with host receptors is mediated by several SU hypervariable regions. In this way, the ASLV viral SU subgroups, A through E are defined.  In the ASLV system, there are numerous strategies and molecular tools to block engagement of viral SU proteins with their cognate receptors including: secreted cell receptor-based immunoadhesin inhibitors, SU immunoadhesins that bind to and block cell receptors, and  superinfection resistance strategies. 

Under such blocking conditions, retroviruses evolve rapidly to expand their host range, and the molecular determinants on the viral SU protein can be mapped. Previously, the authors used a soluble immunoadhesion form of the subgroup A receptor to select for subgroup A viruses with expanded host range.  A deletion in  hr1 resulted in modest expansion of subgroup A usage of B and C receptors, without loss of subgroup A receptor recognition. Here, the authors begin with this  hr1 deletion mutant, and apply more rigorous selection using the comprehensive available tools/strategies with the goal of selecting viruses that have lost subgroup A receptor engagement and have acquired higher affinity for alternative receptors. The authors identify single SU amino acid substitutions that indeed drive selection of such a phenotype. The authors also expand the study to include subgroup C viruses as a starting point. The work is highly comprehensive, the manuscript is very well written, and the conclusions are justified. The discussion of the findings is quite thorough. Moreover, the manuscript will be comfortable for the broad readership of Viruses. 

Minor and stylistic comments:

1. line 48: “variety” is used twice.

2. line 194: “reduced fitness” is perhaps not the best description, as reduced replication is secondary to the toxicity.

3. line 304: add comma: “the results were similar, with a lower level of inhibition”

Author Response

viruses-507879

Reply to Reviewer Report Reviewer 1 (in blue)

Minor and stylistic comments:

1. line 48: “variety” is used twice.

corrected

2. line 194: “reduced fitness” is perhaps not the best description, as reduced replication is secondary to the

toxicity.

edited to state have reduced titers but with no obvious signs of cytotoxicity.

3. line 304: add comma: “the results were similar, with a lower level of inhibition”

corrected

Reviewer 2 Report

In this paper, the authors continue their studies on the evolution and variation in the hypervariable regions of the ASLV Envelope glycoprotein that may explain an expansion of the possible receptors that this virus uses to infect the cells. For this, they are based on their previous studies with mutants Δ155-160 of ASLV-A, and through a strategy of infections in cells that express or not specific types of receptors, they select new viral variants in which they identify mutations of interest in the regions hr1 and v3. The combined studies of the mutations they identify and the profile of receptors that express the cells allow them to propose that certain mutations could give an advantage to viruses, extending the range of surface proteins that allow the entry of viruses into cells.

In this paper the authors should highlight their most significant results in comparison with other previous data, most of their own previous research. The title is too generic since the results shown are in mutants of ASLV-A and ASLV-C and in experiments performed in certain cell lines.

Specific comments:

MATERIAL AND METHODS

Lane 100- A brief description of chicken cell line C and line F15I5, and TF/cks Tva-19 cell line,  must be added (example: C CEFs lack a functional ASLV-A Tva receptor; F15I5 lack a functional ASLV-C Tvc receptor). Also, a general description of the strategies followed to infect different cell lines and obtain viral mutants should be included.

The FAC-based test and the Tva receptors immunoadhesins (Lane 271-273 page 8) should be included in the Materials and Methods section.

RESULTS AND DISCUSSION:

- Figure 2.A. Why the O.D. results in day 2 are so high in two of the inocula?

- Lane 173: authors must eliminate “and G133D+LP43P (1/8)”, as the sentence mentions the predominant clone (G133D+Y142H)

- Lane 198-201: “TF/cksTva-19 cells were also transfected with plasmids encoding the four new mutant viruses, the original RCASBP(A), Δ155-160 mutant virus, the RCASBP(A) wild-type virus, and the RCASBP(B) virus that should not be inhibited by the cksTva-mIgG immunoadhesin. The transfected cell cultures were passaged when confluent to allow virus replication and spread (Figure 3B). As expected, RCASBP(B) replicated well with no growth lag”. Please check figure 3.B because RCASBP (B) data are not shown.

- Lane 260 and 265. Write the name of cells Line 15 CEFs as in Material and Methods

- Lane 335. It is difficult to interpret an inoculum in mL, it should be better in m.o.i.

- Lane 366: authors should explain why some recombinants were constructed only with the F142S mutation if there were no viral mutants with only this additional mutation, and the reason why the data from the experiments with the F142 mutation alone are not shown. In addition, F142S should be highlighted in figure 7.

CONCLUSIONS

The authors should mention any conclusion about the additional vr3 mutations observed in Δ155-160 mutants, since the replication rates of the virus obtained are similar to those of the mutants with additional hr1 mutations (figure 3B and 3C).

FIGURES:

·         Figure 2: Rounds of selection must be added to 2.A and 2.B (similar to figure 6). In the 2.C, it would be better indicate the hypervariable region where the mutations are observed

·         Figure 3: 3.B: The curve for RCASPB (B) is missing, please see text in lanes 198-201. In 3.C: the replication curves of all mutant viruses reach similar values at 16 h pi. Have the authors sequenced the viruses at this time? It seems that independently of the mutation acquired, the viral replication rate is similar in this cell line.

·         Figure 5. Authors must explain why only the representative experiments are shown in this figure.

·         Figure 6. Please, indicate the meaning of “Rd” below the figure itself. In figure 6.D, it would be better indicate the hypervariable region region where the deletion was observed.

Author Response

viruses-507879

Reply to Reviewer Report Reviewer 2 (in blue)

Specific comments:

MATERIAL AND METHODS

Lane 100- A brief description of chicken cell line C and line F15I5, and TF/cks Tva-19 cell line, must be added

(example: C CEFs lack a functional ASLV-A Tva receptor; F15I5 lack a functional ASLV-C Tvc receptor).

The additional information requested has been added.

Also, a general description of the strategies followed to infect different cell lines and obtain viral mutants should

be included.

This approach has been added.

The FAC-based test and the Tva receptors immunoadhesins (Lane 271-273 page 8) should be included in the

Materials and Methods section.

A description of the FACS binding assay and the sources of the ASLV receptor immunoadhesins used in this

study have been added.

RESULTS AND DISCUSSION:

- Figure 2.A. Why the O.D. results in day 2 are so high in two of the inocula?

Virus in supernatant harvested from infected cells was used to infect these cells thus had a high ELISA for the

CA protein at the start of infection. This is in contrast to cells being first transfected with a plasmid encoding a

ASLV where the virus is then initially produced from the plasmid DNA at very low levels, but then spreads and

amplifies through the culture.

- Lane 173: authors must eliminate “and G133D+LP43P (1/8)”, as the sentence mentions the predominant

clone (G133D+Y142H)

Corrected

- Lane 198-201: “TF/cksTva-19 cells were also transfected with plasmids encoding the four new mutant

viruses, the original RCASBP(A), Δ155-160 mutant virus, the RCASBP(A) wild-type virus, and the RCASBP(B)

virus that should not be inhibited by the cksTva-mIgG immunoadhesin. The transfected cell cultures were

passaged when confluent to allow virus replication and spread (Figure 3B). As expected, RCASBP(B)

replicated well with no growth lag”. Please check figure 3.B because RCASBP (B) data are not shown.

We have deleted the lines describing RCASBP(B). This was added in error.

- Lane 260 and 265. Write the name of cells Line 15 CEFs as in Material and Methods

Corrected

- Lane 335. It is difficult to interpret an inoculum in mL, it should be better in m.o.i.

MOI levels were also added.

- Lane 366: authors should explain why some recombinants were constructed only with the F142S mutation if

there were no viral mutants with only this additional mutation, and the reason why the data from the

experiments with the F142 mutation alone are not shown.

The RCASBP(C)+F142S virus was constructed to characterize any phenotype this mutation had on virus

replication apart from the del144-160 mutation. As a result of the RCASBP(C)del144-160 mutant alone virus

not producing virus in DF-1 cells while the RCASBP(C)del144-160+F142S mutant virus being able to replicate,

we hypothesize that the F142S mutation rescues a defect caused by the del144-160 mutation, and that the

escaped mutant virus population from round 3 selection had not completely evolved to all viruses having both

the deletion and F142S mutation. This hypothesis has been added to the discussion of this data.

Reviewer 3 Report

Munguia and Federspiel have previously demonstrated that a sarcoma and leukosis virus of subgroup A (ASLV[A]), in the presence of a competitor of the Tva receptor, evolved to use 

Munguia and Federspiel have previously demonstrated that a sarcoma and leukosis virus of subgroup A (ASLV[A]), in the presence of a competitor of the Tva receptor, evolved to use other host proteins as viral receptors. The selected mutant virus carries a deletion of Env residues 155-160 and exhibits certain ability to use Tvb or Tvc for cell entry.

In this paper, the authors further subject the ASLV(A) delta155-160 virus to genetic selection pressure using a soluble form of the Tva receptor to isolate virus variants that use other cell surface proteins as receptors more efficiently than the parental delta155-160 virus. Moreover, Munguia and Federspiel attempt to obtain an ASLV(C) derivative with the ability to use Tva and/or Tvb proteins as receptors.

In general, the experiments have been thoroughly conducted and the results obtained illustrate how ASLV viruses can accumulate, in a stepwise manner, a series of mutations in the env gene that broaden the repertoire of cell surface proteins that can mediate virus entry.

However, before the manuscript is accepted for publication, the authors should address the following issues:

1- Figure 2A. Since the authors mention in the introduction section that ASLV(A)delta 155-160 was selected by its ability to overcome the block imposed by the Tva inhibitor, it should be expected that this virus replicates efficiently in the TF/cksTva-19 cell line. However, Figure 2 shows that the virus delta155-160 exhibits a delayed replication kinetics. Therefore, it is not clear what advantage the delta155-160 virus has acquired if it still replicates poorly in cells in which usage of Tva is inhibited.

2- Figure 4A. It is shown that delta155-160 virus replicates in DF-1 cells yielding titers higher than 105. By contrast, mutant viruses containing, in addition to the Env deletion, the amino acid substitutions G133D/Y124H, G133D/L143P or G268E replicate in DF-1 cells with an efficiency 10-fold lower than that of the parental delta155-160 virus. Therefore, the 100-fold reduction in the ability of these new mutants to replicate in subgroup B-infected cells with respect to delta155-160 could be partially explained by the 10-fold lower titers that the new mutants del/G133D/Y124H, del/G133D/L143P and del/G268E exhibit in DF-1 cells with respect to that of the delta155-160 virus. The same reasoning can be applied to the impairment in infectivity in subgroup C-infected cells observed for mutants del/G133D/Y124H and del/G133D/L143P. The latter fact is evident when examining the results shown in Figure 4C in which the titers of del/G133D/Y124H, del/G133D/L143P and del/G268E viruses in Line 15 CEFs, which lack the Tvc receptor, are similar to those obtained in DF1 cells. In summary, instead of comparing the titers of the new mutant viruses in cells chronically infected with different subgroup viruses with those of delta155-160, the comparison should be made for each new mutant between its titer in DF-1 and those attained by the same mutant in chronically infected DF-1 cells.

3- Figure 4C. The authors state that mutant virus del/G133D replicates in Line 15 CEFs with an efficiency similar to that of wild-type ASLV(C) virus. To support this conclusion the authors should perform an appropriate statistical analysis including the titers of ASLV(C) in the cell line 15 together with those of all the new mutant viruses obtained after the second genetic selection of ASLV(A)delta155-160.

4- Figure 5. There is a discrepancy between the results showing the binding affinities of the new mutant viruses to chicken Tva and quail Tva immunoadhesins (panels A and C) and those of the effect of these soluble receptors on the ability to infect DF-1 cells (panels B and D). Since the new mutant viruses exhibit low affinity to chicken Tva and undetectable affinity to quail Tva, it is expected that preincubation of these viruses with each immunoadhesin will have a much lesser effect on virus infectivity than that shown in the figure. The authors should provide an explanation for this discrepancy. The conclusion that “all four new mutant viruses were still inhibited with chicken stva-mIgG but at lower levels compared to…” is not substantiated by the data presented in the manuscript.

5- Figure 5 (panels E and F). Only mutant del/G133D/L143P shows a modest affinity to the Tvb receptor representing only 10% of that of ASLV(B) at the maximal concentration of the Tvb receptor (500 ng/ml). Accordingly, the effect of the Tvb immunoadhesin on the infectivity titers of mutant del/G133D/L143P seems to be not statistically significant. Only infectivity of mutant del/G268E appears to be reduced after incubation with the soluble Tvb receptor. This result is unexpected considering that no affinity to the Tvb receptor was detected for this virus. The authors should provide possible explanations for this contradictory result. In addition, the authors should perform a statistical analysis to support the conclusion that previous incubation of mutant viruses del/G133D/L143P and del/G268E with soluble Tvb affects viral infectivity in DF-1 cells.

6- Based on the observation that mutant ASLV(C)del144-160/F142S replicates inefficiently in DF-1 cells chronically infected with subgroup A, B and C viruses, it would be highly important to provide the results of an experiment aimed at determining the affinity binding of this virus to Tva, Tvb and Tvc receptors. Without these new data, the phenotype of ASLV(C)del144-160/F142S remains poorly characterized. Alternatively, the authors may decide to exclude from the manuscript the preliminary results obtained when attempting to select ASLV(C) variants with broadened tropism and show them elsewhere, together with additional experiments. 

Author Response

viruses-507879

Reply to Reviewer Report Reviewer 3 (in blue)

However, before the manuscript is accepted for publication, the authors should

address the following issues:

1- Figure 2A. Since the authors mention in the introduction section that ASLV(A)delta 155-160 was selected by

its ability to overcome the block imposed by the Tva inhibitor, it should be expected that this virus replicates

efficiently in the TF/cksTva-19 cell line. However, Figure 2 shows that the virus delta155-160 exhibits a

delayed replication kinetics. Therefore, it is not clear what advantage the delta155-160 virus has acquired if it

still replicates poorly in cells in which usage of Tva is inhibited.

As already described in lines 124-135, there is a major difference between the antiviral effect of the SUA-rIgG

inhibitor that will bind and down-regulate Tva receptors requiring mutations in ASLV(A) to use an alternative

receptor, and the antiviral effect of the sTva-mIgG inhibitor that binds directly to the ASLV(A) glycoprotein

trimers preventing infection requiring mutations to reduce the binding affinity of the sTva-mIgG inhibitor to the

trimer. While the TF/chsTva-19 cells express normal levels of the cellular Tva receptor, the high levels of the

secreted sTva-mIgG inhibitor produced by the cell line not only out competes the membrane-bound Tva

receptor, but upon binding to the ASLV(A) virion trimers, directly prevents the virus from using an alternative

receptor for entry until dislodging the sTva-mIgG inhibitor. Therefore, the delta 155-160 virus has no

advantage in the presence of the sTva-mIgG inhibitor since it still binds with high affinity.

2- Figure 4A. It is shown that delta155-160 virus replicates in DF-1 cells yielding titers higher than 105. By

contrast, mutant viruses containing, in addition to the Env deletion, the amino acid substitutions G133D/Y124H,

G133D/L143P or G268E replicate in DF-1 cells with an efficiency 10-fold lower than that of the parental

delta155-160 virus. Therefore, the 100-fold reduction in the ability of these new mutants to replicate in

subgroup B-infected cells with respect to delta155-160 could be partially explained by the 10-fold lower titers

that the new mutants del/G133D/Y124H, del/G133D/L143P and del/G268E exhibit in DF-1 cells with respect to

that of the delta155-160 virus.

With all due respect, we do not agree with this reviewers reasoning on how to interpret the receptor

interference assays. Also, we are just measuring titers 2-days after infection not performing growth replication

studies.

The titer of each virus was determined on uninfected DF-1 cells and at the same time the same virus stock was

titered on A-, B- C- and J-preinfected DF-1 cells. As expected, maximum infectious titers are achieved on

uninfected DF-1 cells and J-preinfected cells since subgroup J is not homologous.

First: the initial observation was comparing the titers on uninfected DF-1 cells:

Wild-type ASLV(A) ~106 ifu/mL

(A)delta155-160 ~105 ifu/mL

(A)delta+new mutations all ~104 ifu/mL

So the 'cost' to the wild-type (A) virus evolving to (A)delta155-160 mutant and expanding receptor usage was

~10-fold of the maximum titer.

In this manuscript, we demonstrate the 'cost' to the (A)delta155-160 mutant to block sTva-mIgG inhibitor

binding and further expand receptor usage was an additional ~10-fold of the maximum titer compared to the

(A)delta155-160 mutant, and 100-fold compared to wild-type (A) virus..

Second: the pattern of titers on the preinfected cells was compared to titers on uninfected DF-1 cells.

Wild-type ASLV(A) >105-fold reduction on (A)

(A)delta155-160 ~103-fold reduction on (A); ~2-5-fold reduction on (B) and (C)

(A)delta+G133D ~100-fold reduction on (A); ~20-fold reduction (B) and (C)

(A)delta+GD+YH ~100-fold reduction on (A), (B) and (C)

(A)delta+GD+LP ~100-fold reduction on (A), (B) and (C)

(A)delta+G268E ~100-fold reduction on (A), (B) and (C)

Round  2

Reviewer 3 Report

With all due respect to the authors, I am not entirely convinced by their response to my criticisms of their experimental data. However, they have clarified somewhat some of the questions I raised in my review.

Besides, since the other two reviewers made positive comments on the paper, I consider that the manuscript could now be accepted for publication.